# Intravenous Immunoglobulins at the Crossroad of Autoimmunity and Viral Infections

**DOI:** 10.3390/microorganisms9010121

**Published:** 2021-01-07

**Authors:** Carlo Perricone, Paola Triggianese, Roberto Bursi, Giacomo Cafaro, Elena Bartoloni, Maria Sole Chimenti, Roberto Gerli, Roberto Perricone

**Affiliations:** 1Rheumatology, Department of Medicine, University of Perugia, 06129 Perugia, Italy; carlo.perricone@gmail.com (C.P.); dott.roberto.bursi@gmail.com (R.B.); giacomo.cafaro@libero.it (G.C.); elena.bartolonibocci@unipg.it (E.B.); roberto.gerli@unipg.it (R.G.); 2Rheumatology, Allergology and Clinical Immunology, Department of “Medicina dei Sistemi”, University of Rome, 00133 Rome, Italy; maria.sole.chimenti@uniroma2.it (M.S.C.); roberto.perricone@uniroma2.it (R.P.)

**Keywords:** autoimmunity, infection, intravenous immunoglobulins, SARS-CoV-2, virus

## Abstract

Intravenous immunoglobulins (IVIG) are blood preparations pooled from the plasma of donors that have been first employed as replacement therapy in immunodeficiency. IVIG interact at multiple levels with the different components of the immune system and exert their activity against infections. Passive immunotherapy includes convalescent plasma from subjects who have recovered from infection, hyperimmune globulin formulations with a high titer of neutralizing antibodies, and monoclonal antibodies (mAbs). IVIG are used for the prevention and treatment of several infections, especially in immunocompromised patients, or in case of a poorly responsive immune system. The evolution of IVIG from a source of passive immunity to a powerful immunomodulatory/anti-inflammatory agent results in extensive applications in autoimmune diseases. IVIG composition depends on the antibodies of the donor population and the alterations of protein structure due to the processing of plasma. The anti-viral and anti-inflammatory activity of IVIG has led us to think that they may represent a useful therapeutic tool even in COVID-19. The human origin of IVIG carries specific criticalities including risks of blood products, supply, and elevated costs. IVIG can be useful in critically ill patients, as well as early empirical treatment. To date, the need for further well-designed studies stating protocols and the efficacy/tolerability profile of IVIG and convalescent plasma in selected situations are awaited.

## 1. Introduction

Intravenous immunoglobulins (IVIG) are a blood preparation pooled from the plasma of tens of thousands of donors who underwent plasmapheresis in order to obtain a very high concentration of immunoglobulins (Ig). The most abundant component of IVIG is IgG, usually over 95%, although other fractions of Ig can be present as well. Various IVIG formulations are commercially available and differ in terms of concentration, Ig content (Ig isotypes, IgG, IgM, IgA, and IgE), and other characteristics [1,2].

IVIG were first employed to treat immunodeficiencies in the 1950s and are currently licensed for the treatment of primary immunodeficiencies (PIDs) with impaired antibody production, secondary immunodeficiencies with recurrent infections, antibody deficiency, or proven specific antibody failure (PSAF) [3]. Following the first evidence on the ability of IVIG to reduce the rate of infections in immunocompromised patients, reports have been published suggesting beneficial effects in idiopathic thrombocytopenic purpura (ITP). Data have been accumulated on several other autoimmune diseases and inflammatory conditions supporting the current license of IVIG treatment in patients with ITP, Guillain-Barré syndrome, Kawasaki disease, chronic inflammatory demyelinating disease, and multifocal motor neuropathy. Beyond their licensed indications, the off-label use of IVIG is very common as well, in clinical practice, in specific conditions including severe inflammatory myopathies, myastenia gravis, Lambert-Eaton myastenic syndrome, Lyell’s syndrome, and others. However, because of the rarity of most of these diseases, useful large studies and randomized clinical trials (RCTs) are awaited [3,4,5].

IVIG contain structurally and functionally intact Ig with half-lives of approximately 18–32 days, (similar to native IgG) and a normal proportion of subclasses: 95% monomeric IgG, small amounts of dimers, variable amounts of IgA and IgM. IVIG may also contain traces of soluble molecules including human leukocyte antigen (HLA) and cytokines. They do not contain high molecular weight immune complexes and contaminants such as vasomotor peptides and endotoxins. IVIG are subjected to industrial manipulation, processes of inactivation, and chemical and physical removal of bacteria and viruses. IVIG products may differ in their pharmaceutical properties (osmolality, pH, sodium content, stabilizer, IgA content) which can affect safety and tolerability [6].

## 2. Mechanisms of Immunomodulatory Action

IVIG interact at multiple levels with the different components of the immune system [7,8]. Indeed, beyond the antigen-specific effects, IVIG exert anti-inflammatory activity through the interactions between the Fc domain of the IgG and their receptors (FcγRs) [9]. In autoimmune diseases characterized by thrombocytopenia mediated by the presence of antiplatelet antibodies, IVIG probably induce the blockade of FcγRs present on monocytes and macrophages involved in the mechanism of antibody-dependent cytotoxicity and phagocytosis of autoantibody-coated platelets. The Fc-dependent immunomodulation exerted by IVIG can also involve alternative cellular mechanisms. In fact, IVIG induce the expression of FcγRIIB inhibitory receptors with consequent inhibition of B-cell activation and induction of anergy and/or apoptosis through the phosphorylation of intracellular immunoreceptor tyrosine-based inhibitory motif (ITIM) domains. Another proposed mechanism of action concerns the saturation of “neonatal” Fc receptors (FcRn), which involves an accelerated clearance of circulating pathogenic antibodies [10]. IVIG also allow the neutralization of circulating autoantibodies through the anti-idiotype antibodies contained in them: the interaction between IVIG and B-cell receptor idiotypes is at the basis of the ability of IVIG to regulate self-reactive B cell clones in vivo. In fact, IVIG are able to induce a block in the proliferation of B cells: this seems to be due to the differentiation of a subset of non-proliferating IgG secreting B cells [11]. IVIG exert their immunosuppressive effect at the level of dendritic cells, inhibiting the expression of HLA and CD80/CD86 molecules as well as the production of interleukin (IL)-2 [11]. It has been postulated that the modulation effect of the cytokine network, with the induction of the production of anti-inflammatory cytokines (such as IL-10), represents the main anti-inflammatory action of IVIG in vivo [12]. IVIG interfere with the activated complement components and the formation of the membrane attack complex (MAC) through the binding of C3b and C4b and, consequently, avoiding the interaction of these components with cellular targets [13]. Natural anti-C3b antibodies have been revealed able to inhibit the activity of C5 convertase in vitro [13]

Another anti-inflammatory mechanism of IVIG seems to be mediated by the pool of IgG with α-2,6 sialylated terminal residues (sFc) at Fc [14]. These sFc binds the dendritic cell-specific intercellular adhesion molecule-3-grabbing non-integrin (DC-SIGN) receptor on the cell surface of macrophages and the mouse counterpart SIGN-R1 (specific intracellular adhesion molecule-grabbing non-integrin R1). Through these interactions, sFc promotes the expression of cytokines and immunosuppressive receptors (FcγRIIB). According to this model, the anti-inflammatory effects of IVIG treatment are weaker in the knockout mouse model for SIGN-R1 but can be restored in the knock-in model with human DC-SIGN. Studies suggest the presence of other surface lectins as alternative ligands for the Fc fragment of IgG [8,15]. Immunomodulatory activities of IVIG also include the Tregitope “(regulatory T-cell epitopes present in IgG)” mechanism in tolerance induction mainly through the modulation of the regulatory T-cell axis, Tregs-cytokine expression, reduction of IL-17, and enhancement of the suppressive function of Tregs [16]. IVIG can promote the suppressive function of the T regulatory cells, probably thanks to the mediation of dendritic cells [17,18,19,20]. In addition, it has been shown that they have an inhibitory effect on the differentiation and amplification of T helper (Th)17 cells, reducing inflammatory cytokines and other pro-inflammatory mediators’ production, thereby interfering with the maintenance of chronic inflammatory response [21]. IVIG can induce apoptosis of B and T cells through the activation of Fas receptor, are able to block the binding between T cells and superantigens, to control self-reactivity and induce tolerance, and to inhibit the differentiation and maturation of dendritic cells.

## 3. Therapeutic Indications in Rheumatology

The Food and Drug Administration (FDA) has approved the use of IVIG for the treatment of PIDs since the 1980s. The therapeutic goal in these conditions is to reach a serum IgG level in the recipient > 5 g/dL, before the next infusion, although this threshold value and its potential “protective” effect is still debated [22]. The recommended dose is 400–600 mg/kg approximately every 2–4 weeks, with an inter-infusion interval that varies from patient to patient. The therapeutic benefits of IVIG in this context were initially traced to the ability to deliver specific antibodies to recipients unable to produce them-particularly antibodies to encapsulated microorganisms such as *Streptococcus pneumoniae* and *Haemophilus influenzae*. However, simultaneously, data emerged on IVIG efficacy in improving autoimmune hematological complications (such as ITP) [23]. Subsequent studies showed that high doses of IVIG were able to provide a significant rise in the platelet count and often a resolution of clinical features in children with ITP. Extensive use of IVIG was thus promoted in other autoimmune diseases. 

To date, IVIG are used in the treatment of a broad spectrum of severe rheumatological diseases including dermatomyositis (DM) and systemic vasculitides, such as KD, ANCA-positive small vessels vasculitides (AAV) [24,25,26,27,28]. In this context, several studies support the efficacy of high-dose IVIG in the therapy of eosinophilic granulomatosis with polyangiitis (EGPA), mainly in patients with neurological and/or cardiovascular complications [29,30].

IVIG treatment is also applied in clinical practice in systemic lupus erythematosus (SLE), polymyositis (PM), anti-phospholipid antibody syndrome (APS), and others [5]. In such pathological conditions, the IVIG dosage is usually 2 g/kg administered over 2–5 consecutive days.

IVIG are currently used for the treatment of patients with severe SLE who do not respond to other immunosuppressive drugs or as a steroid-sparing agent, mainly in patients with lupus nephritis [30]. Case reports and open-label trials describe that high-dose IVIG are effective in improving numerous clinical manifestations in SLE patients including neuro-SLE [31]. IVIG can represent a first-line therapy in cases of neuro-SLE, in patients who are not candidates for other immunosuppressants such as cyclophosphamide, or in patients with concomitant infections [31,32]. Many therapeutic interventions are available to treat patients with PM and DM including corticosteroids, immunosuppressive drugs, and plasmapheresis: evidence documents that IVIG represent an efficacious therapy replacing or reducing steroid and immunosuppressive medications in PM/DM patients, mainly in induction for refractory cases or when immunosuppressive drugs are contraindicated [33].

IVIG represent a key treatment to manage APS refractory to conventional therapy [34,35]. Furthermore, in pregnant women with primary or secondary APS to SLE, various therapeutic protocols with IVIG have been successfully applied over the years with safety and efficacy on mothers and newborns [36,37,38,39].

## 4. Anti-Viral Aspects of IVIG 

IVIG is a well-known treatment for a variety of diseases not only as a replacement therapy but also for its efficacy against infections and its anti-inflammatory and immune-modulating effects in autoimmune disorders [40,41,42]. The anti-viral effects of Ig include their activity in preventing cell penetration and activating innate immune system cells and the complement pathways [42]. Preparations of IVIG that are obtained from healthy donors contain various polyclonal Ig directed against a wide variety of antigens.

IVIG are currently used for the prevention and treatment of several infections, especially in immunocompromised patients, or in the case of severe and poorly responsive autoimmune disease (Table 1) [42]. The administration of IVIG finds indication in infections of subjects with an impaired immune system or as a substitute treatment for patients with hypogammaglobulinemia (primary or secondary deficiency) to prevent or treat common opportunistic viral and bacterial infections [43,44,45]. IVIG composition depends on the antibody composition of the donor population [46,47]. However, IVIG usually present significant activities against different viruses, like cytomegalovirus (CMV), varicella-zoster virus (VZV), herpes simplex virus (HSV), hepatitis A virus (HAV), respiratory syncytial viral (RSV), Epstein-Barr virus (EBV), measles, mumps, rubella, parvovirus B19 [48], and polyomavirus BK [49]. IVIG may also be effective in treating drug-resistant or severe CMV, parvovirus B19, and polyomavirus BK infections in post-transplant patients [50,51]. Several case reports have described the successful use of IVIG in the treatment of anemia caused by chronic parvovirus B19 infection. IVIG therapy has been shown to clear viremia and to improve symptoms and cytokine dysregulation in parvovirus B19–associated chronic fatigue. Parvovirus B19 infection is highly prevalent in the general population, IVIG contain a significant anti–parvovirus B19 concentration and are considered the only specific treatment of the viral infection.

IVIG, especially the so-called “hyperimmune preparations”, i.e., Ig collected from donors with high titers of desired antibodies, are still used for the treatment of a variety of infectious diseases and infection-related disorders (botulism, CMV, hepatitis B, rabies, tetanus) [39,52]. In transplant patients, the use of IVIG (400 mg/kg on days 1, 2, and 7 and 200 mg/kg on day 14) combined with antiviral drugs such as acyclovir and gancyclovir, seems to prevent CMV related complications, such as pneumonitis, whereas either treatment alone does not [52,53,54]. In the meanwhile, the same combination does not seem to give benefit in case of CMV gastrointestinal involvement [55,56]. IVIG were also used as prophylaxis for VZV infection in newborns exposed to the virus after birth and were effective for the treatment of a disseminated VZV infection [57,58]. In addition, IVIG seem to reduce the recurrence of genital manifestations of HSV-2 [59]. Although they are efficacious in reducing the risk of secondary infection in HIV-infected children, they have no efficacy against HIV infection and should not be considered as antiviral therapy in HIV patients [39].

Among the viruses to which IVIG may exert defensive activity, there is also HAV: as reported, standard immunoglobulins preparations may be utilized in selected, susceptible patients for the prevention of HAV [60,61]. In a comparative study between HAV vaccine and IVIG for post-exposure prophylaxis, no significant difference, rather a slightly higher IVIG-induced protection, was observed [62].

Few data on immunodeficient patients treated with ribavirin combined with IVIG (500 mg/kg every other day) for the treatment of RSV pneumonitis are available. Survival rates in these series compared with those expected based on historical cohorts were encouraging, and suggested a benefit from IVIG as an adjunct therapy to ribavirin [63,64]. However, even a humanized monoclonal antibody has been approved and is available for immunodeficient patients [65,66,67].

Evidence documents the use of IVIG in EBV-related diseases including the prevention of EBV-associated post-transplant lymphoproliferative disease (PTLD), in association with acyclovir or ganciclovir [68], as well as in conservative treatment without etoposide in patients with hemophagocytic lymphohistiocytosis (EBV-HLH) [69]. Moreover, IVIG and plasma exchange are the standard therapy for acute inflammatory demyelinating polyneuropathy such as Guillain-Barré syndrome that has been linked to EBV, CMV, and other viruses [70].

## 5. IVIG in COVID-19

During the current Coronavirus disease 2019 (COVID-19) outbreak, IVIG have been used in different clinical settings with the objective to evaluate their efficacy against severe forms [42]. Among various pathways by which the severe acute respiratory syndrome coronavirus 2 (SARS-CoV-2) can trigger inflammation and tissue damage, the Fc receptor-mediated antibody-dependent enhancement (ADE) has been suggested and may occur when antiviral neutralizing antibodies cannot completely neutralize the virus [42,71]. Fu et al. hypothesized that in the absence of a proven clinical FcR blocker, the use of IVIG could be a viable option for the urgent treatment of pulmonary inflammation to prevent lung injury; in fact, IVIG could be used to saturate the IgG recycling capacity of FcRn and proportionately reduce the levels of antiviral neutralizing antibodies [72].

A multicenter, double-blind, randomized controlled trial for cases with severe influenza A (H1N1) infection demonstrated that hyperimmune Ig (H-IG) can reduce the serum concentration of cytokines such as interferon (IFN)-γ, IL-12, soluble tumor necrosis factor receptor 1 (sTNFR1), CXCL10, CXCL9, CCL3, viral load, and have an impact on mortality rates [73]. Interestingly, currently available IVIG seem to contain antibodies reacting against SARS-CoV-2 and that can be explained as a cross-reaction with the common circulating coronaviruses [74]. This allows us to hypothesize that in an analogy with what has been observed for influenza A (H1N1), even in COVID-19, IVIG may prevent the SARS-CoV-2-induced lung damage by reducing the cytokine release [42].

Numerous case-reports describing the effectiveness of IVIG against SARS-CoV-2 are available in the literature; however, it is quite hard to collect strong evidence from them, because of the different IVIG preparations and doses, patients’ comorbidities, and standards of care. Several authors suggested that IVIG were able to improve radiological findings and viral clearance in patients with COVID-19-related interstitial multifocal pneumonia, within a few days of clinical remission [75,76]. Evidence of successful usage, in terms of clinical and radiological healing, of IgM-enriched polyclonal IVIG, administered at the dose of 5 mg/kg/daily for 3 days, for a time of 12 h, in addition to the standard of care, has been reported [75]. It has been documented that severe cases of SARS-CoV-2 were successfully treated without mechanical ventilation or intensive supportive care by using intensive plasma exchange along with IVIG. Similarly, IVIG given at a dose of 0.3–0.5 g/kg continuously for five days were able to improve the clinical condition and O_2_ saturation in COVID-19 patients, thereby preventing the progression of pulmonary lesions [76]. IVIG showed to be a useful weapon against SARS-CoV-2, especially in transplant patients, where there is a high risk of drug interactions [77]. High-dose IVIG therapy enhances passive immunity and has a modulating effect. Therefore, IVIG can be a therapeutic option in the early stages of SARS-CoV-2 infection and seem to reduce the need for mechanical ventilation and to promote early recovery improving the outcome as well as disease course [78,79,80,81]. The use of IVIG in COVID-19 patients also gave good results in some cases of meningoencephalitis and in multiple cases of Guillain-Barrè syndrome in Italy and in other countries [82,83,84,85]. IVIG have also been used in cases of Evans’ syndrome and ITP secondary to COVID-19 [86,87,88]. Pouletty et al. described 16 cases of KD-like disease following SARS-CoV-2 infection treated with IVIG, but only five patients (31%) were successfully treated with a single IVIG infusion, while 10 patients (62%) required a second treatment [89]. Anecdotal cases described the use of IVIG in acute disseminated encephalomyelitis and pemphigoid after SARS-CoV-2 infection [90,91].

Currently, studies to evaluate the efficacy and safety of IVIG in combination with standard care for severe COVID-19 exist but further RCTs are required [92], thus caution is needed.

Ig specific serum antibodies could be associated with an increased risk of infection by promoting viral entry in susceptible cells through a mechanism called antibody-dependent enhancement (ADE), firstly described for Flaviviridae, HIV, and Ebola viruses [93]. Through ADE, the virus-neutralizing antibodies complex can attach to FcR of the Ig, facilitating the viral endocytosis and infection of the target cells and consequently leading to an increase of viral replication and to greater disease severity. This interaction results in the upregulation and release of the pro-inflammatory cytokines (such as MCP-1 and IL-8), macrophage (M1 or classically activated) responses, and their accumulation in lungs with consequent lung injury [93]. Furthermore, the use of IVIG can lead to two potential and severe adverse effects: the transfusion (immunoglobulin)-related acute lung injury (TRALI), an immune-mediated process that can manifest as acute respiratory distress, and the thrombotic events related to IVIG treatment (incidence of 1–16.9%) [94].

## 6. Passive Immunotherapy and COVID-19: IVIG, Convalescent Plasma, Hyperimmune Immunoglobulins, and Monoclonal Antibodies

Passive immunotherapy includes convalescent plasma (CP) from subjects who have recovered from an infection, purified formulations created by the pharmaceutical industry with a high titer of neutralizing antibodies (hyperimmune globulin, H-IG), and monoclonal antibodies (mAbs) [95,96,97,98,99,100,101]. In this context, serotherapy constitutes the most important form of passive immunity while H-IG have been used as a second line passive immunotherapy against infectious diseases. The transfer of pathogen-specific Ig isotypes (IgG, IgM, IgA, and IgE) and innate humoral immune factors is the challenge of passive immunity; a high neutralizing antibody titer (dose adjusted per recipient body weight) is required for therapeutic effectiveness [102]. The CP has a wide immunomodulatory action on several molecules including anti-inflammatory cytokines and coagulation factors leading to an improvement of the severe inflammatory response triggered by the viral infection [103]. Such passive immunotherapy is able to exert both active inflammatory cytokines neutralization and passive viral neutralization, thus inducing immunomodulation, by reduction of inflammation, inflammation-associated injury, and viral load. The CP conferred immunity is short-term and the effectiveness of passive immunotherapy is higher if administered for preventive purposes or at an early stage. Both CP and H-IG depend on plasma collected from patients who have recovered from the diseases [104]. Hyperimmune plasma represents a further use of CP since it is a polyclonal H-IG concentrate obtained through apheresis from a pool of plasma from patients who recovered from an infectious disease and who have developed high serum titers of neutralizing antibodies [101]. H-IG is a drug derived from CP (usually 10% final concentration) that is produced following normal Ig production processes resulting in a purified mix of polyclonal Ig against target proteins, with a standardized titer. H-IG is pharmacokinetically like IVIG, suggesting that a single infusion of H-IG may be sufficient in acute disease settings [105]. It is essential to note that CP and H-IG should be collected at least 3 weeks post-coronavirus diseases, including COVID-19, to increase the chances of obtaining high-titer neutralizing antibodies because of the median seroconversion time for IgM and IgG, that is, 12 and 14 days from disease onset, respectively [106]. At day 15 from disease onset, the presence of antibodies was 94.3% for IgM and 79.8% for IgG. Interestingly, in patients with COVID-19, it has been observed an initial antibody response with “early” antibodies, mainly anti-Spike protein (Ab-S), and a delayed antibody response with “late” antibodies, mainly anti-nucleocapsid (Ab-N) with a neutralizing action [107,108]. Moreover, the antibody levels in convalescent patients have been related to the initial viral load and independently associated with the disease severity in middle east respiratory syndrome (MERS) [109].

Studies on severe acute respiratory syndrome (SARS) or severe influenza globally reported that passive immunotherapy was able to decrease the mortality of patients, especially with early treatment [110]. In SARS caused by SARS-CoV-2, CP or H-IG, which is highly homologous to SARS-CoV-2, have been reported to reduce mortality and hospital stay [111] (Table 2). Duan et al. described that severe COVID-19 recipients of CP with high-titer neutralizing activity from individuals who had recovered from COVID-19 improved clinically with a rapid increase in serum neutralizing antibody titers and no detectable SARS-CoV-2 viral RNA in their blood [112]. Results from COVID-19 suggest that the administration of IVIG and CP is safe, reduces the viral load, and improves the clinical course [77,78,79,113]. Moreover, in the management of severe COVID-19 patients, passive immunotherapies also prevent intensive care unit admission and mechanical ventilation, both of which are limited resources [114,115]. In SARS-CoV-2 infection, the evidence that both high antibody titers and early seroconversion are associated with worse outcomes supports that the timing of highly active CP transfusion may be critical [102,116]. Timely administration is thus strongly recommended, with optimal outcomes in the first 7 days, a good efficacy within 14 days, and certainly no indications beyond three weeks after the onset of the disease. Thus, patients who have rapidly progressing yet early disease may benefit from the antibody boost coming from H-IG. The timing and therapeutic dosage to be used according to clinical experiences are 250–300 mL of hyperimmune plasma administered to each of the admitted patients for a maximum of 3 times within 5 days [117]. Based on the SARS-CoV-2 studies, therapeutic hyperimmune plasma was used at a dosage of 5 mL/kg with an Ab neutralizing titer of 1: 160 [118]. Other indications for COVID-19 H-IG also include pre-exposure prophylaxis in front-line health care providers (HCPs). However, to date, no high-quality data have demonstrated convincingly the efficacy and safety of H-IG and CP as treatments for coronavirus diseases including COVID-19 [119]. On 24 March 2020, the FDA of the United States published guidelines for the use of the CP−COVID-19 and defined three pathways for therapeutic access to CP: 1. compassionate use; 2. clinical trials; 3. therapy protocols. The use for prophylactic purposes is not allowed [120,121]. As stated in the recommendations, CP has to be obtained from an FDA-registered blood establishment that follows the donor eligibility criteria and qualifications in collecting plasma: donors must have had documented infection with SARS-CoV-2 (RT-PCR or serology) and be at least 28 days from the resolution of symptoms or at least 14 days without symptoms, and a negative RT-PCR test [122]. Requirements for anti-COVID-19 hyperimmune plasma donor candidates have been highly defined along with specific eligibility criteria. Apheresis is the recommended procedure for obtaining plasma with an effectiveness about 400–800 mL of plasma from a single donor [122,123,124]. In addition, FDA does not collect COVID-19 CP or provide COVID-19 CP: the HCP or acute care facilities should instead obtain COVID-19 CP from an FDA-registered blood establishment.

Recent results from studies on a large population of critically ill patients show that CP is potentially effective and sufficiently safe in hospitalized patients with COVID-19. Several clinical trials are ongoing to assess safety, efficacy, appropriate effective dose, and dose-response of CP transfusion in COVID-19 patients and to specifically define clinical and laboratory improvements after CP [125,126,127,128,129]. However, no definitive demonstration of a beneficial effect is available, as recently reported [119]. Rather, a RCT on CP in patients with severe COVID-19-associated pneumonia did not allow us to observe any significant difference between the patients who had received CP compared with those who had received placebo at day 30 from the start of treatment [130].

Limitations of CP and H-IG include 1. screening for blood-borne pathogen (CP must be negative for the viral markers HBV, HIV, HCV serology and molecular, Syphilis serology; HAV and Parvovirus B19-DNA (<105 copies/mL) tests must also be performed); 2. blood type matching; 3. the batch-to-batch variability including purity, and the content of IgG, IgM, IgA, and the specific antibody titer (by enzyme immunoassay-EIA must be >160 or equivalent by another method). As a remark, the different qualities of antibodies in CP results at different stages of the disease. For this reason, both a proper evaluation of the antibody response in CP and an adequate titer of neutralizing SARS-CoV-2 antibodies present in the collected plasma are essential [123,124]. The H-IG are pre-validated for their neutralizing activity; thus, they do not pose the dilemma of validating the neutralizing activity as with the CP. Finally, CP and H-IG are subject to all precautions regarding human plasma therapies including, in particular, the caution against circulatory overload and the absolute contraindication of use in subjects with complete IgA deficit. Furthermore, a note of caution when treating COVID-19 patients with CP is the potential risk of ADE likely linked to early seroconversion prior to virus clearance with an enhancement of macrophage-dependent inflammatory damage by non-neutralizing antibodies. The tendency to ADE could be genetically determined thus, susceptible patients could develop worsening paradoxical symptomatology [126].

## 7. Conclusions 

A summary of the hypothetical mechanisms by which IVIG may exert beneficial or detrimental effects in COVID-19 is shown in Figure 1.

Antibody-based immunotherapies including IVIG, CP, and H-IG have proven efficacy and safety and have been used for decades [131]. They are already being used and found to be useful to manage COVID-19 patients too, even though a definitive demonstration of effectiveness is still lacking [132,133,134].

The extensive and long-standing use of IVIG is widely documented and advantages/disadvantages of IVIG-therapies have been described (Table 3). One of the main advantages of IVIG is the safety profile: few adverse effects have been reported and most of them are mild, usually not relapsing, and well manageable [6]. Immediate reactions to IVIG administrations are the most common and mainly associated with rapid infusion: they consist of malaise, headache, myalgia, arthralgia, nausea, fever, and chest discomfort. Late adverse events mainly include acute renal failure (associated with sucrose preparations) and thrombotic events (related to excessive doses or fast IVIG infusions and the content of coagulation factors). Both events can be prevented by an adequate choice of patients for IVIG therapy (patients with multiple risk factors) and by using good hydration and slow infusion rates, monitoring kidney function, and avoiding sucrose-containing preparations [6].

The disadvantages of IVIG are mainly related to their human origin. IVIG are blood products prepared from large human plasma pools and that brings specific criticalities: the risks associated with blood products (e.g., infections, even though this risk is currently negligible, infusion adverse reaction), the sufficiency of their supply, and the elevated costs. To date, the potential for transmission of infection to recipients has been dramatically reduced by the standards of donor care and new procurement technologies with the implementation of effective viral inactivation procedures into the production process of all IVIG preparations. Thus, the possibility of infection transmission is negligible. 

**Table 1 microorganisms-09-00121-t001:** Passive immunotherapy in viral infectious diseases.

Viruses	IVIG	Ref.	H-IG	Ref.	mAb	Ref.
Herpesviruses						
HSV	X	[48,59]				
VZV	X	[48,57,58]	X	[4]		
CMV	X	[50,52,53,54,55,56]	X	[4]		
EBV	X	[48]				
Hepatitis A virus	X	[60,61]				
RSV	X	[63,64]			X	[4]
Measles	X	[48,60]				
Mumps	X	[48]				
Rubella	X	[48]				
Parvovirus B19	X	[48,50,60]				
Polyomavirus BK	X	[49,50]				

IVIG, intravenous immunoglobulins; H-IG, hyperimmune globulin; mAb, monoclonal antibodies; HSV, Herpes Simplex Virus; VZV, Varicella Zoster Virus; CMV, cytomegalovirus; EBV, Epstein-Barr virus; RSV, respiratory syncytial virus.

**Table 2 microorganisms-09-00121-t002:** Passive immunotherapy in COVID-19.

Passive Immunotherapy	Definition	Evidence in COVID-19 (Ref.)
**IVIG**	Purified IgG from a pool of thousands of donors	[42,75,76,77,78,79,80,81]
**CP**	Whole plasma from convalescent donors containing specific antibodies at titer ≥ 1:160	[97,98,112,113,114,115,116,117,118,119,127,130,131]
**H-IG**	IVIG obtained from a pool of plasma with high titer of specific antibodies	[97,105]
**mAbs**	Fully human, neutralizing mAbs against SARS-CoV-2 spike protein	[42,133]

IVIG, intravenous immunoglobulins; CP, convalescent plasma; H-IG, hyperimmune globulin; mAbs, monoclonal antibodies; SARS-CoV-2, severe acute respiratory syndrome coronavirus 2.

**Table 3 microorganisms-09-00121-t003:** Advantages and disadvantages of IVIG therapy in infectious diseases.

Advantages	Disadvantages
History of efficacy	Proven clinical utility (lack of RCTs)
High compliance and better disease control (repeated hospital access)	Less flexibility for patients and parents (infusion center/hospital)
Useful in critical patients	High costs (products, supply, nurses)
High dosage therapy	Larger volume
Safety profile (common)	Fatal systemic adverse events (rare)
Useful in patients with bleeding disorders	Risk of thrombosis (rare)
Useful in pregnancy	Need intravenous access
Rapid achievement of plasma levels and response (<72 h)	Transient response (<1 month)
Early empiric therapy (before the identification of pathogens)	Antibody-mediated enhancement (ADE)

In conclusion, IVIG use has rapidly grown in treating a variety of autoimmune diseases and infections including COVID-19. In addition, evidence from COVID-19 and previous pandemics documents that the use of CP had a favorable safety profile with no adverse events reported among these patients [133]. In this context, an unmet need is to establish standardized universally accepted protocols (timing, duration, optimal dose) since an important hurdle to overcome is the high cost of IVIG and CP that may be partially justified by the reduction of disease complications requiring costly hospitalizations. Future studies would also benefit from the characterization of patient antibody levels prior to CP transfusion and examination of anti-viral IgA and IgM in both patient and donor plasma should also be considered, as all three classes and their subclasses have different roles in naturally occurring immune responses.

## Figures and Tables

**Figure 1 microorganisms-09-00121-f001:**
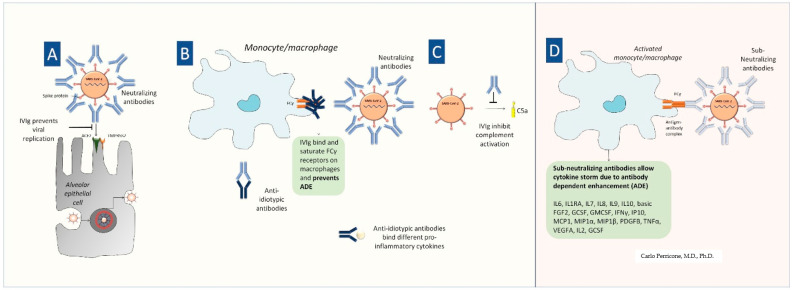
The hypothetical dual role of IVIG in SARS-CoV-2 infection. IVIG can be beneficial since (**A**) neutralizing antibodies binding spike protein may prevent SARS-CoV-2 attachment to the ACE2 receptor thus inhibiting viral entry into the cell and in turn viral replication; (**B**) IVIG can bind and saturate Fcγ receptors on innate immune cells (e.g., macrophages) in the lungs, thus preventing ADE. Anti-idiotypic antibodies can bind to anti-viral antibodies as well as to proinflammatory cytokines. (**C**) IVIG are well-known modulators of the complement cascade activated by SARS-CoV-2; (**D**) Fc receptor-mediated ADE may occur when antiviral neutralizing antibodies cannot completely neutralize the virus thus macrophages (and other innate immune cells) are stimulated towards the exaggerated inflammatory response leading to the cytokine storm. [**A**, **B**, **C**, and **D** in the figure refer to the same letters in the above-mentioned comments in the text.].

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
