# Peer review of "Intravenous Immunoglobulins at the Crossroad of Autoimmunity and Viral Infections"

_microorganisms, 2021, doi:10.3390/microorganisms9010121_

Round 1

Reviewer 1 Report

A good review about the use of IVIG product and its use for infectious diseases

ADE is mentioned at different part of your paper, even in conclusions, but, in spite of the fact it has been mentioned for SARS , ADE has been only suggested for SARS-CoV-2 so, currently, it cannot entirely be ruled out, further research should be made at any case. I think this information should be shown in the paper.

Regarding convalescent plasma and derived products (IVIG) batches, theses can be very consistents, since a surveillance of antibody titers in plasma donors is performed.

In regards to other concerns or disadvantages, the fact to infuse IVIG in the hospital is not a big disadvantage for COVID-19 patients, since patients who need the treatment (as far as I know) are usually hospitalized. Maybe this point should be qualified.

And, of course, currently plasma derived proteins for therapeutic uses are a very safe products, the posibility of infection transmision is negligible

.

Author Response

We thank Reviewer #1 for the useful comments on our manuscript. 

  • As Reviewer #1 highlighted, the ADE has been only suggested for SARS-CoV-2: in the revised version of the paper, we added details on this information quoting Tetro JA in Microbes Infect. 2020 Mar;22(2):72-73. 
  • Regarding IVIG disadvantages, we highlighted, in conclusions, that to infuse IVIG in the hospital is not a big disadvantage for COVID-19 patients, since patients who need the treatment are usually hospitalized. 
  • In the revised version of the paper, we added additional consideration on IVIG safety (the possibility of infection transmission is negligible).

Reviewer 2 Report

Perricone et al provide a manuscript on the role for intravenous immunoglobulins (IVIG) for autoimmunity and viral infections for this journal entitled Microorganisms.  The authors provide much information on IVIG and did a lot of work.  For this journal, there is too much discussion of the immunomodulatory treatment aspects of IVIG. In discussing immunomodulation by Ig, the authors should familiarize themselves with Tregitopes, discovered by Anne De Groot and Bill Martin at EpiVax.  The discussion would be helped with comments about monoclonal antibody therapies.   Also, the text is disorganized and very repetitive, as if individual authors wrote sections and they were not edited into a single document.  For example, the  immunomodulatory aspect of IVIG was repeated several times in the antiviral section. The conclusion section brings in too much new information and is too long.  At least 2 references were mis-numbered or inappropriate.  Much of the text was phrased awkwardly, illogically presented, or lacked focus.  For this journal, this manuscript would be better served by focusing on the antiviral activities of IVIG and related products with a short comment on its immunomodulatory therapeutic action, possibly tying it to immunomodulation in Covid 19.  There were many issues in the manuscript, some specific comments follow:

  1. 2 par 1. Line 4 what specific conditions?

       Par 2. Line 2. Mention should be made here of different IgG subtypes.

       Par 3. Line 3 Runon sentence

                  Line 8 What is meant by ‘inhibition of cellular B activation’ ?

                  Lines 11 and 15 need references.

       Par 4.  Is this the only mechanisms of anti-inflammatory activity?

  1. 3 par 1. Lacks focus

                par 2. Too much detail on treatment for a review. What is the point?  Paragraph is repetitive and needs to be reorganized.

                                Line 8. Meaning of ‘Grade A evidence of effectiveness’

  1. 4. Par 1. Repeats the treatment of autoimmune diseases rather than focusing on antiviral aspects. Needs an introduction to basic biology of how antibody can be antiviral and follow up on that story. Also, generic ability of IVIG to elicit protections based on immune exposures of the population vs hyperimmune IG.  The immunomodulatory action is secondary to the main theme of antiviral activity and more appropriate as a special aspect of anti-Covid 19. 

                Par 2. Repetitive.

                                Line 12: typo: poliomavirus BK

                                Line 16: which virus?

                                Line 18: Start a new paragraph to distinguish hyperimmune preparations e.g. specific antibody treatments.

                                References 57 and 59 are incorrect.

                                This paragraph provides a laundry list of possible uses of IVIG but is should be clear that they are not valid treatments.

  1. 5. The list of IVIG applications for viral infections would make a good table including whether it is an established or anecdotal report.

                Section 5. Needs focus and less repetition

                Par 1. What is Nab?

                Par 2.    Paragraph is difficult to read due to awkward phrasing, runon sentences and illogical presentation.

Line 1  ?’outspread’

Line 6.  IVIG may have antibodies to SARS. Why?  ? Cross reactivity towards common cold coronavirus?  If so, cite references.

                Par 3.  This paragraph has too much detail on treatment

Line2 Reference?

  1. 6. Par 1. Repetitive.

                What is ADE? Adverse drug effects?

                Has ADE been shown for Coronaviruses? If not, then this issue should be presented in the earlier antiviral paragraph.

                Section 6. repetitive and disorganized.

  1. 7 par 2. Line 21 (starts with However……) contradicts previous statements.
  2. 8. Figure legend. Explain A, B, C, D. if only to say that the letters refer to comments in the text.

                Also, the cytokine box in D does not relate to ADE. 

                Par 2 needs more explanation of the statements.

                Par 4 is repetitive

Author Response

We thank Reviewer #2 for the useful comments on our manuscript. 

Reviewer #2: In discussing immunomodulation by Ig, the authors should familiarize themselves with Tregitopes, discovered by Anne De Groot and Bill Martin at EpiVax.  The discussion would be helped with comments about monoclonal antibody therapies. Also, the text is disorganized and very repetitive, as if individual authors wrote sections and they were not edited into a single document.  For example, the immunomodulatory aspect of IVIG was repeated several times in the antiviral section. The conclusion section brings in too much new information and is too long.  At least 2 references were mis-numbered or inappropriate.  Much of the text was phrased awkwardly, illogically presented, or lacked focus.  For this journal, this manuscript would be better served by focusing on the antiviral activities of IVIG and related products with a short comment on its immunomodulatory therapeutic action, possibly tying it to immunomodulation in Covid 19.

Authors: We deeply revised the paper considering all the comments by reviewers and Academic Editor. In the revised version, we introduced Tregitopes in discussing immunomodulation by IVIG. We also removed several redundant concepts to avoid confusion. We rearranged the references and also quoted additional evidence from the literature, as suggested by reviewers and Academic Editor.

Reviewer #2: There were many issues in the manuscript, some specific comments follow:

    • 2 par 1. Line 4 what specific conditions? = REVISED;
    • Par 2. Line 2. Mention should be made here of different IgG subtypes = REVISED;
    • Par 3. Line 3 Runon sentence = UNCLEAR;
    • Line 8 What is meant by ‘inhibition of cellular B activation’ ? + Lines 11 and 15 need references. = REVISED;
    • Par 4.  Is this the only mechanisms of anti-inflammatory activity? = Several anti-inflammatory activities have been discussed in the paper;
    • Lacks focus. Too much detail on treatment for a review. What is the point?  Paragraph is repetitive and needs to be reorganized. = In the revised paper, we reorganized several points among the paragraphs and also removed excessive details on treatments in rheumatologic diseases;
    • Line 8. Meaning of ‘Grade A evidence of effectiveness’ = REMOVED;
    • Repeats the treatment of autoimmune diseases rather than focusing on antiviral aspects. Needs an introduction to basic biology of how antibody can be antiviral and follow up on that story. Also, generic ability of IVIG to elicit protections based on immune exposures of the population vs hyperimmune IG.  The immunomodulatory action is secondary to the main theme of antiviral activity and more appropriate as a special aspect of anti-Covid 19. = REVISED
    • Par 2. Repetitive. = REVISED;
    • Line 12: typo: poliomavirus BK = REVISED;
    • Line 16: which virus? = REVISED;
    • Line 18: Start a new paragraph to distinguish hyperimmune preparations e.g. specific antibody treatments. = REVISED;
    • references 57 and 59 are incorrect. = REVISED;
    • This paragraph provides a laundry list of possible uses of IVIG but is should be clear that they are not valid treatments = The overview of possible uses of IVIG goes from evidence in clinical practice in both the prophylaxis and the treatment to the potential applications in specific conditions relying on relevant reports from the literature. However, the debated efficacy of several IVIG protocols has been discussed. 
    • The list of IVIG applications for viral infections would make a good table including whether it is an established or anecdotal report. = As suggested by Academic Editor, we deleted anecdotal reports, in the revised paper.
    • Section 5. Needs focus and less repetition = The revised version of the paper has been streamlined and simplified;
    • Par 1. What is Nab? = REVISED;
    • Par 2.    Paragraph is difficult to read due to awkward phrasing, runon sentences and illogical presentation. = REVISED;
    • Line 1  ?’outspread’ = REVISED;
    • Line 6.  IVIG may have antibodies to SARS. Why?  ? Cross reactivity towards common cold coronavirus?  If so, cite references. = REVISED;
    • Par 3.  This paragraph has too much detail on treatment, Line2 Reference? = REVISED.
    • What is ADE? Adverse drug effects? Has ADE been shown for Coronaviruses? If not, then this issue should be presented in the earlier antiviral paragraph. = REVISED (details on antibody-dependent enhancement (ADE) have been added);
    • Section 6. repetitive and disorganized. 7 par 2. Line 21 (starts with However……) contradicts previous statements. = REVISED;
    • Figure legend. = REVISED;
    • Par 2 needs more explanation of the statements and Par 4 is repetitive = The revised version of the paper has been streamlined and simplified.

Round 2

Reviewer 2 Report

The revised manuscript from Perricone on the role for intravenous immunoglobulins (IVIG) for autoimmunity and viral infections for Microorganisms remains repetitive, confusing and disorganized with many sentences that are run on or awkwardly written. The reader should not have to wait until the end of the manuscript to learn of the different types of passive immunity to distinguish IVIG,  High titer gammaglobulin, plasma and high titer plasma and their role in viral therapy. Each should be defined at the beginning with their roles and uses in antiviral therapy.  The list of viruses treated with these therapies should be in the beginning to validate the manuscript.  As a microbiologist, the discussion of immunomodulatory activity is an interesting sideline but should not be the major thrust, especially for this journal, and the discussion can follow sections on antiviral action unless there is more on the mechanisms of antiviral activity (including a discussion of ADE).  Also, is there a negative concern for immunomodulation with IVIG for antiviral activity, can it be immunosuppressive of an essential protection? If not, say so. 

Possibly the major contribution of this manuscript is the discussion of anti-coronavirus activities and this is not even mentioned in the Abstract.  

The conclusion section is not a conclusion because it is too long and introduces new concepts such as a whole new section on treatments during pregnancy, unrelated to antiviral action.

Without a complete revision, this manuscript is not acceptable.

It would be helpful if the lines were numbered within the text.

SPECIFIC COMMENTS

Abstract: 3rd to last line. Increase in plasma levels of what? No mention of coronavirus sections.  This is probably the most important part of the manuscript.

P4. Mechanism of action section should discuss antiviral mechanisms including potential ADE.  Then some discussion of immunomodulation.  The detail for immunomodulation is repeated in the rheumatology section and maybe should be combined into that section.   Also, the relevant question is whether immunomodulation is good or bad for antiviral therapy.

               Line 16. What is the consequence of the FcRn action? Does IVIG promote or inhibit Ig clearance? Is this good or bad?

P5. Line 5.  The  sFc  is not the only mechanism for anti-inflammatory action but this sentence implies that it is.

Section 3.  This section repeats previous section and is very disorganized. What relevance does dosing details for autoimmunity have? for antiviral actions?   Recommend either deleting entire section or combine with immunomodulation as described above.

               8th line from bottom: new sentence is a runon sentence and awkwardly constructed.

Section 4. This section is disorganized. Need to relate all the statements to anti viral activity.  Otherwise, this section is just a restatement of previous sections.   As mentioned, this should be the main theme of the manuscript and be introduced earlier. The first 7 lines of this section repeat previous.  

  1. 9. Need to distinguish early in the manuscript the difference between generic IVIG and post disease plasma and pooled antibodies, mcAb therapies. Can repeat that IVIG is likely to contain Ig against normal microbial exposures. Also, hyperimmune gammaglobulin and plasma.  A table of the different forms of passive immunity would be good.

            Line 6 Discussion of the limitations of IVIG due to manufacturing was mentioned earlier and repetition is not necessary.

            4th line from bottom.  What type of IVIG is this? 

p.10      Line 6. How does this relate to antiviral?

            Line 14. GAMMAGLOBULIN was standard treatment for HAV, rabies and other virus infections for many years. Present a List of passive immunization for viruses would be appropriate.

            10th from bottom. Antibody treatments for RSV in infants?

The section on ADE is confusing.  Is antibody good or bad?  Is IVIG good or bad?

p.15 Why is a discussion of MERS thrown in here?

            9th line from bottom.  Why is H-IG discussed in the context of CP? Change in nomenclature?

p.16 line 5. RT-PCR for what?  Also, should be screened for HIV, HBVetc as mentioned later.

Author Response

Reviewer: The revised manuscript from Perricone on the role for intravenous immunoglobulins (IVIG) for autoimmunity and viral infections for Microorganisms remains repetitive, confusing and disorganized with many sentences that are run on or awkwardly written. The reader should not have to wait until the end of the manuscript to learn of the different types of passive immunity to distinguish IVIG, High titer gammaglobulin, plasma and high titer plasma and their role in viral therapy. Each should be defined at the beginning with their roles and uses in antiviral therapy.  The list of viruses treated with these therapies should be in the beginning to validate the manuscript.  As a microbiologist, the discussion of immunomodulatory activity is an interesting sideline but should not be the major thrust, especially for this journal, and the discussion can follow sections on antiviral action unless there is more on the mechanisms of antiviral activity (including a discussion of ADE).  Also, is there a negative concern for immunomodulation with IVIG for antiviral activity, can it be immunosuppressive of an essential protection? If not, say so.

Authors: We are thankful with the Reviewer for his arguments that we tried to fully address his considerations in the revised version of the manuscript. In accordance with the Reviewer, we removed repetitive sentences and reorganized several parts of the manuscript. We highlighted the importance of different types of passive immunity by adding tables on IVIG, high titer gammaglobulin, convalescent plasma and the role in viral therapy, including COVID-19. In addition, comments on immunomodulatory activity of IVIG have been revised by discussing all its facets in a single paragraph.

Reviewer: Possibly the major contribution of this manuscript is the discussion of anti-coronavirus activities and this is not even mentioned in the Abstract. 

Authors: According to the Reviewer, we mentioned, in the revised manuscript, the anti-coronavirus activities in the Abstract.

Reviewer: The conclusion section is not a conclusion because it is too long and introduces new concepts such as a whole new section on treatments during pregnancy, unrelated to antiviral action.

Authors: According to the Reviewer, we deeply revised the conclusions’ session by adequately removing new concepts from this session and by adding many of them in other parts of the text (e.g. pregnancy, manufacturing).

Reviewer: Without a complete revision, this manuscript is not acceptable.

Authors: We tried to address all the arguments of the Reviewer with a deep revision of the whole text, newly quoted references, and additional tables, in order to make the paper suitable for publication.

Reviewer: It would be helpful if the lines were numbered within the text.

Authors: Accordingly, lines have been numbered within the text.

SPECIFIC COMMENTS

Reviewer: Abstract: 3rd to last line. Increase in plasma levels of what? No mention of coronavirus sections.  This is probably the most important part of the manuscript.

Authors: Accordingly, the text of the abstract has been modified in the revised text.

Reviewer: P4. Mechanism of action section should discuss antiviral mechanisms including potential ADE.  Then some discussion of immunomodulation.  The detail for immunomodulation is repeated in the rheumatology section and maybe should be combined into that section.  Also, the relevant question is whether immunomodulation is good or bad for antiviral therapy.

Authors: According to the Reviewer, we deeply revised the paragraph on immunomodulation and anti-viral therapy and we adequately modified the rheumatology section.

Reviewer: Line 16. What is the consequence of the FcRn action? Does IVIG promote or inhibit Ig clearance? Is this good or bad?

Authors: The revised manuscript discussed these arguments. You can read: “Another proposed mechanism of action concerns the saturation of “neonatal” Fc receptors (FcRn) which involves an accelerated clearance of circulating pathogenic antibodies [10].”

Reviewer: P5. Line 5.  The sFc is not the only mechanism for anti-inflammatory action but this sentence implies that it is.

Authors: According to the Reviewer, we modified the sentence as follows: “Another anti-inflammatory mechanism of IVIG seems to be mediated by the pool of IgG with α-2,6 sialylated terminal residues (sFc) at Fc [14]”.

Reviewer: Section 3.  This section repeats previous section and is very disorganized. What relevance does dosing details for autoimmunity have? for antiviral actions? Recommend either deleting entire section or combine with immunomodulation as described above.

Authors: According to the Reviewer, in the new version of the manuscript, we deeply revised the section 3.

Reviewer: 8th line from bottom: new sentence is a runon sentence and awkwardly constructed.

Authors: According to the Reviewer, we modified the final part of the section 3 in order to make the text more clear.

Reviewer: section 4. This section is disorganized. Need to relate all the statements to anti viral activity.  Otherwise, this section is just a restatement of previous sections. As mentioned, this should be the main theme of the manuscript and be introduced earlier. The first 7 lines of this section repeat previous. 

Authors: According to the Reviewer, in the new version of the manuscript, we deeply revised the section 4.

Reviewer: 9. Need to distinguish early in the manuscript the difference between generic IVIG and post disease plasma and pooled antibodies, mcAb therapies. Can repeat that IVIG is likely to contain Ig against normal microbial exposures. Also, hyperimmune gammaglobulin and plasma.  A table of the different forms of passive immunity would be good.

Authors: In revised version of the review, two additional tables on  IVIG and post disease plasma and pooled antibodies, mcAb have been included.

Reviewer: Line 6 Discussion of the limitations of IVIG due to manufacturing was mentioned earlier and repetition is not necessary.

Authors: According to the Reviewer, in the new version of the manuscript, we modified this specific part of the discussion.

Reviewer: 4th line from bottom.  What type of IVIG is this? p.10      Line 6. How does this relate to antiviral?

Authors: According to the Reviewer, in the new version of the manuscript, we adequately changed the text.

Reviewer: Line 14. GAMMAGLOBULIN was standard treatment for HAV, rabies and other virus infections for many years. Present a List of passive immunization for viruses would be appropriate.

Authors: According to the Reviewer, in the new version of the manuscript, we included an addition table on that topic.

Reviewer:10th from bottom. Antibody treatments for RSV in infants?

Authors: Antibody treatments for RSV are on use in infants (evidence from Pediatrics has been quoted)

Reviewer: The section on ADE is confusing.  Is antibody good or bad?  Is IVIG good or bad?

Authors: The mechanism of ADE has not been clearly reported for IVIG; however, when it occurs, it’s a “bad mechanism” because it improve the viral strategies becoming a mechanism of “viral escape”.

Reviewer: p.15 Why is a discussion of MERS thrown in here?

Authors: MERS has been mentioned at this level in accordance with the subject matter.

Reviewer: 9th line from bottom.  Why is H-IG discussed in the context of CP? Change in nomenclature?

Authors: According to the Reviewer, in the new version of the manuscript, we revised the text.

Reviewer: p.16 line 5. RT-PCR for what?  Also, should be screened for HIV, HBV etc as mentioned later.

Authors: RT-PCR is for SARS-CoV-2.
